# Proteomic Analysis Reveals Differential Expression Profiles in Idiopathic Pulmonary Fibrosis Cell Lines

**DOI:** 10.3390/ijms23095032

**Published:** 2022-05-01

**Authors:** Juan Manuel Velázquez-Enríquez, Alma Aurora Ramírez-Hernández, Luis Manuel Sánchez Navarro, Itayetzi Reyes-Avendaño, Karina González-García, Cristian Jiménez-Martínez, Luis Castro-Sánchez, Xariss Miryam Sánchez-Chino, Verónica Rocío Vásquez-Garzón, Rafael Baltiérrez-Hoyos

**Affiliations:** 1Laboratorio de Fibrosis y Cáncer, Facultad de Medicina y Cirugía, Universidad Autónoma “Benito Juárez” de Oaxaca, Oaxaca 68120, Mexico; juanmanuelvela_enriquez@live.com (J.M.V.-E.); aramih_09@hotmail.com (A.A.R.-H.); itayetzi.reyes94@gmail.com (I.R.-A.); k.igg@hotmail.com (K.G.-G.); 2Facultad de Medicina y Cirugía, Universidad Autónoma “Benito Juárez” de Oaxaca, Oaxaca 68020, Mexico; sluismanuel81@hotmail.com; 3Departamento de Ingeniería Bioquímica, Escuela Nacional de Ciencias Biológicas, Instituto Politécnico Nacional, Unidad Profesional Adolfo López Mateos, Zacatenco, Av. Wilfrido Massieu Esq. Cda. Miguel Stampa S/N, Alcaldía Gustavo A. Madero, Mexico City 07738, Mexico; cjimenezh@ipn.mx; 4Conacyt-Centro Universitario de Investigaciones Biomédicas “CUIB”, Universidad de Colima, Colima 28045, Mexico; luis_castro@ucol.mx; 5Catedra-Conacyt, Departamento de Salud El Colegio de La Frontera Sur, Unidad Villahermosa, Tabasco 86280, Mexico; xsanchez@ecosur.mx; 6Conacyt-Facultad de Medicina y Cirugía, Universidad Autónoma “Benito Juárez” de Oaxaca, Oaxaca 68120, Mexico; veronicavasgar@gmail.com

**Keywords:** idiopathic pulmonary fibrosis, fibroblasts, proteomic analysis, mass spectrometry, proteomics, differentially expressed proteins, KEGG pathway

## Abstract

Idiopathic pulmonary fibrosis (IPF) is a chronic, progressive, irreversible lung disorder of unknown cause. This disease is characterized by profibrotic activation of resident pulmonary fibroblasts resulting in aberrant deposition of extracellular matrix (ECM) proteins. However, although much is known about the pathophysiology of IPF, the cellular and molecular processes that occur and allow aberrant fibroblast activation remain an unmet need. To explore the differentially expressed proteins (DEPs) associated with aberrant activation of these fibroblasts, we used the IPF lung fibroblast cell lines LL97A (IPF-1) and LL29 (IPF-2), compared to the normal lung fibroblast cell line CCD19Lu (NL-1). Protein samples were quantified and identified using a label-free quantitative proteomic analysis approach by liquid chromatography-tandem mass spectrometry (LC-MS/MS). DEPs were identified after pairwise comparison, including all experimental groups. Gene Ontology (GO) enrichment analysis, Kyoto Encyclopedia of Genes and Genomes (KEGG), and Protein–Protein Interaction (PPI) network construction were used to interpret the proteomic data. Eighty proteins expressed exclusively in the IPF-1 and IPF-2 clusters were identified. In addition, 19 proteins were identified up-regulated in IPF-1 and 10 in IPF-2; 10 proteins were down-regulated in IPF-1 and 2 in IPF-2 when compared to the NL-1 proteome. Using the search tool for retrieval of interacting genes/proteins (STRING) software, a PPI network was constructed between the DEPs and the 80 proteins expressed exclusively in the IPF-2 and IPF-1 clusters, containing 115 nodes and 136 edges. The 10 hub proteins present in the IPP network were identified using the CytoHubba plugin of the Cytoscape software. GO and KEGG pathway analyses showed that the hub proteins were mainly related to cell adhesion, integrin binding, and hematopoietic cell lineage. Our results provide relevant information on DEPs present in IPF lung fibroblast cell lines when compared to the normal lung fibroblast cell line that could play a key role during IPF pathogenesis.

## 1. Introduction

Idiopathic pulmonary fibrosis (IPF) is a chronic, progressive, and irreversible lung disorder of unknown cause, associated with a histopathological and radiological pattern of usual interstitial pneumonia (UIP), occurring most frequently in adult males over 60 years of age [1,2]. The high burden of symptoms and comorbidities has a critical impact on the patient’s quality of life, characterized by a decline in lung function leading to death from respiratory failure within approximately 3 to 5 years of diagnosis [3,4]. However, despite the severity of IPF, the basis of the molecular mechanisms that promote the development of the disease remains an unmet need, resulting in limited or inadequate treatment options [2,5]. Within these mechanisms, profibrotic activation of resident lung fibroblasts resulting in aberrant accumulation of extracellular matrix (ECM) proteins is described as a hallmark of IPF [5,6].

The presence of fibroblastic foci (FF) is one of the characteristics of the lungs of patients with IPF and represent aggregates of activated fibroblasts that migrate to injury areas, maintain a high rate of proliferation, and secrete an excessive amount of ECM proteins, thus contributing to the abnormal remodeling of the lung architecture [5,7,8]. The aberrant and persistent activation of resident lung fibroblasts during the development of IPF is regulated by several soluble factors and alterations present in the lung microenvironment, such as transforming growth factor β (TGF-β), recognized as a critical player in the development of fibrosis [8,9,10]. In this sense, it has been evidenced that IPF lung fibroblasts show profibrotic properties such as a high proliferation rate, resistance to apoptosis, and an increased capacity for ECM synthesis. Likewise, it has been demonstrated that these fibroblasts retain these profibrotic properties when maintained under in vitro conditions [6,11]. Furthermore, recent research has shown that soluble factors released by IPF lung fibroblasts act on normal lung fibroblasts and promote various fibrogenesis-related processes [12]. Therefore, exploring protein expression and function in IPF lung fibroblasts would significantly explain the cellular and molecular mechanisms associated with IPF progression.

In the last decade, proteomic tools have enabled high-throughput studies to identify protein expression in different chronic diseases [13,14]. Therefore, the use of proteomics has facilitated the identification of specific diagnostic biomarkers, new therapeutic targets, and critical proteins related to the main molecular mechanisms responsible for developing a specific disease [13,15]. In this regard, a significant number of investigations have focused on the application of proteomic approaches to study the differential proteomic profile in different samples from IPF patients, such as lung tissue, serum, and bronchoalveolar lavage fluid (BALF) [15,16,17,18,19,20]; However, limited studies have aimed to identify the proteomic profile of IPF fibroblast cell lines.

In the present study, we performed a label-free quantitative proteomic analysis by liquid chromatography-tandem mass spectrometry (LC-MS/MS) of IPF lung fibroblasts compared to normal lung fibroblasts (NL), to gain insight into the profiles of differentially expressed proteins (DEPs). Furthermore, we employed a bioinformatics approach to describe the pathways, biological processes (BPs), molecular functions (MFs), and cellular components (CCs) related to these proteins that might be involved in the pathogenesis of IPF. Therefore, this work aimed to identify DEPs and provide information on the various cellular and molecular pathways activated in IPF fibroblasts. These results provide relevant information that would contribute to the discovery of new potential therapeutic targets to stop the development of IPF, and candidate biomarkers of the disease, and provide a foundation that helps to understand fibroblasts role in the development of IPF.

## 2. Results

### 2.1. The Workflow of Proteomic Analysis

We adopted label-free quantitative proteomic analysis to investigate the DEPs between the LL29 (IPF-1), LL97A (IPF-2), and CCD19Lu (NL-1) cells. The experimental workflow of proteomics studies is shown in Figure 1, as explained in detail in Methods. In brief, we acquired LC-MS/MS data on a Q Exactive HF mass spectrometer. The raw mass spectrometry (MS) files were analyzed and searched in the UniProt human protein database using the Maxquant platform. Proteins recording a label-free quantification (LFQ) intensity value ≠ 0 in at least one out of six total samples and with at least two MS/MS spectral counts were considered for further analysis, identifying 1832 proteins (Appendix A).

### 2.2. Reproducibility of LC-MS/MS Data

We used LFQ intensity values to quantify the relative abundance of proteins for each group of cell lines. Consistency of LFQ intensity values is critical for accurately measuring the abundance of proteins identified in multiple samples. Figure 2A presents the box plot for the log2 values of the LFQ intensity, corresponding to the duplicate samples of each group, indicating that the interquartile range and median are similar between the duplicates of each group, showing the consistency of the values obtained in the LC-MS/MS measurements. In addition, we obtained a high coefficient of determination (R2) value for the LFQ intensities (Figure 2B). The R2 values between replicates for the proteome of each cell line group were: 0.98 for NL-1, 0.99 for IPF-1, and 0.97 for IPF-2. These data indicate a higher correlation between biological replicates in each group than between replicates from different samples.

### 2.3. Distribution of Proteins Identified by Proteomic Analysis

We then used the LFQ intensity expression value and the MS/MS spectral count to classify the 1832 proteins into the three groups of cell lines. A protein was considered present in a group only when the LFQ intensity value was different from zero, and the MS/MS was equal to two or higher in at least one of the two biological replicates. Figure 2C shows the Venn diagram distribution of each group’s unique and common proteins. The analysis showed 1584 proteins in NL-1, 1669 in IPF-1, and 1725 in IPF-2, of which 1497 proteins were common in all groups. Unique proteins were 15 in number for NL-1, 58 in IPF-1, and 110 in IPF-2 (Appendix A). In addition, we identified 80 proteins exclusively from the fibrotic groups (IPF-1 and IPF-2) (Appendix A).

### 2.4. Gene Ontology (GO) and Kyoto Encyclopedia of Genes and Genomes (KEGG) Analysis of Proteins Identified Exclusively in the IPF-2 and IPF-1 Groups

We performed GO and KEGG pathway analysis to better understand the functions and pathways in which the proteins identified exclusively in the IPF-1 and IPF-2 groups (80 proteins) may be involved. The results revealed that these proteins are associated with BP such as DNA damage response, signal transduction by p53 class mediator, glutathione derivative biosynthetic process, regulation of endocytosis, and cell cycle (Figure 3A). Besides, concerning MF, these proteins are mainly involved in protein binding, motor activity, N-acetylgalactosamine-4-sulfatase activity, and glutathione transferase activity (Figure 3B). Moreover, in connection with the CC, proteins were enriched for extracellular exosome, membrane, focal adhesion, and cytosol (Figure 3C). In addition, the results obtained from the KEGG pathway enrichment analysis revealed that these proteins are related to pathways such as glutathione metabolism and lysosome (Figure 3D).

### 2.5. Identification of DEPs in IPF Cell Lines

We used the LFQ intensity values to calculate the DEPs in two pairwise comparison groups: IPF-1 vs. NL-1 and IPF-2 vs. NL-1. First, we determined DEPs’ fold change (FC) values, obtained by subtracting the mean of the log2 values (Δlog2 (LFQ intensity)) of all standard and single proteins in both groups. Proteins that showed an FC ≥ 1.5 but ≤ −1.5 and a *p*-value < 0.05 were considered significant. Volcano plots show DEPs between FPI-1 vs. NL-1 (Figure 4A) and IPF-2 vs. NL-1 (Figure 4B). Results indicate that 19 and 10 proteins for IPF-1 and IPF-2, respectively, are up-regulated (≥1.5 and *p* < 0.05), as well as 10 and 2 proteins, for IPF-1 and IPF-2, respectively, are down-regulated (≤−1.5 and *p* < 0.05) compared to the NL-1 proteome (Appendix A). In addition, the up-regulated proteins identified in both comparison groups equaled a number of six (Figure 4C). On the other hand, we identified no down-regulated proteins in both comparison groups (Figure 4D).

### 2.6. GO and KEGG Analysis of DEPs

We performed GO and KEGG pathway analysis to better understand the functions and pathways in which DEPs may be involved (Figure 5). Results show significantly up-regulated proteins are associated with BP, such as cell adhesion, angiogenesis, actin filament coating, and cytoskeleton organization (Figure 5A). In contrast, significantly down-regulated proteins are associated with mitochondrial ATP synthesis coupled with proton transport, negative apoptotic process regulation, precursor metabolites, and energy generation (Figure 5E). On MF, it was observed that the up-regulated proteins are related to integrin binding, ankyrin binding, and cadherin binding involved in cell-cell adhesion (Figure 5B); the analysis did not identify significant terms for MF related to down-regulated proteins. Furthermore, up-regulated proteins were associated with CC, such as extracellular exosome, myelin sheath, extracellular vesicle, and focal adhesion (Figure 5C). In contrast, down-regulated proteins were enriched in the mitochondrion, extracellular exosome, myelin sheath, and mitochondrial matrix (Figure 5F). KEGG pathway enrichment analysis showed that these up-regulated proteins were related to pathways such as metabolic pathways (Figure 5D), and the down-regulated proteins were mainly enriched in ways such as metabolic pathways and the citrate cycle (TCA cycle) (Figure 5G).

### 2.7. Protein–Protein Interaction (PPI) Network Construction and Analysis of Hub Proteins

Initially, we evaluated the interaction of the 80 proteins identified exclusively in the fibrosis groups (IPF-1 and IPF-2) using the search tool for retrieval of interacting genes/proteins (STRING), which provided us with a PPI network consisting of 80 nodes and 49 edges, with a mean node degree of 1.23 and a clustering coefficient of 0.41. The expected number of edges was 28, indicating a much lower value than the actual edges encountered, and the *p*-value of the PPI enrichment was 0.000296 (Appendix A). Likewise, the STRING PPI network analysis of the DEPs showed a tight PPI network, suggesting that the proteins have strong interactions; the analysis revealed that this PPI network contains 35 nodes and 34 edges, with an average node degree of 1.94 and a clustering coefficient of 0.547. The expected number of edges was 10, indicating a much lower value than the actual edges encountered, and the *p*-value of the PPI enrichment was 1.75 × 10^−9^ (Appendix A).

In addition, we evaluated the interaction of DEPs and 80 proteins identified exclusively in the fibrosis groups (IPF-1 and IPF-2) and merged it to understand the relationship and the interaction between these proteins to predict their involvement in IPF development. The analysis was performed using the STRING online tool, and the PPI network was visualized with Cytoscape software. The results indicate that the proteins are involved in a complex interaction network containing 115 nodes and 136 edges, with an average node degree of 2.37 and a clustering coefficient of 0.436. The expected number of edges was 73, indicating a much lower value than the actual edges found, and the *p*-value of PPI enrichment was < 3.09 × 10^−11^ (Figure 6A).

Subsequently, we identified the top 10 hub proteins of the PPI network by the Maximum Clique Centrality (MCC) method using the CytoHubba add-on of Cytoscape software. The results showed that the top 10 hub proteins were Activated Leukocyte Cell Adhesion Molecule (ALCAM), CD9 antigen (CD9), Thy-1 Membrane Glycoprotein (THY1), Endoglin (ENG), Alanyl Aminopeptidase (ANPEP), 5’-Nucleotidase Ecto (NT5E), Integrin alpha-3 (ITGA3), ADP-ribosylation factor GTPase-activating protein 1 (ARFGAP1), Spectrin Alpha Chain, Non-Erythrocytic 1 (SPTAN1), and Spectrin beta chain, non-erythrocytic 1 (SPTBN1) (Figure 6B).

Finally, we performed GO and KEGG analysis for these proteins to better understand the functions and pathways in which the 10 hub proteins are related (Table 1). The analysis showed that the hub proteins were significantly enriched for BP, such as cell adhesion, positive regulation of GTPase activity, ER to Golgi vesicle-mediated transport, and cytoskeleton organization. In the case of MF, terms such as integrin-binding were observed to be enriched. In addition, concerning CC, showed that they were related to terms such as the external side of the plasma membrane, focal adhesion, extracellular exosome, and spectrin. Moreover, the results obtained from the KEGG pathway analysis indicated that three proteins (ITGA3, ANPEP, CD9) were significantly related to the hematopoietic cell lineage pathway.

## 3. Discussion

IPF is a devastating disease with a high mortality rate and a median life expectancy of 3 to 5 years after diagnosis [4]. Until now, the pathophysiological mechanism promoting the development of IPF remains an unmet need [2,5]. Aberrant activation of fibroblasts plays an essential role in developing this disease [8]. These fibroblasts are characterized by maintaining a high capacity for proliferation and resistance to apoptosis [11]. Also, they keep an increased expression of α-smooth muscle actin (α-SMA) and secretion of ECM components such as fibronectin and collagen [21]. Lung fibroblasts isolated from IPF patients maintained these profibrotic characteristics under in vitro culture conditions [2,21]. Therefore, the identification of proteins differentially expressed by cultured IPF lung fibroblasts would provide salient data on the main pathophysiological mechanisms related to the role of fibroblasts in IPF progression.

In this study, label-free quantitative proteomic analysis was used to identify DEPs in the proteome of LL97A (IPF-1) and LL29 (IPF-2) fibroblast cell lines bearing a human IPF phenotype when compared to the proteome of the CCD19Lu (NL-1) fibroblast cell line bearing a normal human lung phenotype. We identified eighty proteins unique to IPF-1 and IPF-2. In addition, a total of 29 differentially expressed identified proteins in IPF-1 vs. NL-1 and 12 in IPF-2 versus NL-1. STRING analysis showed that DEPs exclusively expressed in the fibrosis group compose a complex IPP network. In addition, the MCC method identified the 10 hub proteins with the highest degree of connectivity, including ENG, THY1, CD9, NT5E, ITGA3, SPTBN1, SPTAN1, ALCAM, ANPEP, and ARFGAP1. GO analysis showed that these hub proteins were enriched in BP as cell adhesion and associated with MF and CC as integrin binding and focal adhesion. Moreover, KEGG pathway analysis showed three hub proteins (ITGA3, ANPEP, CD9) were significantly enriched for the pathway hematopoietic cell lineage.

It is important to note that some of these hub proteins have been studied because of their close relationship with the pathogenesis of various fibrotic diseases. For example, ENG, also called CD105, is a membrane glycoprotein that acts as a type III helper receptor for the TGFβ superfamily and promotes the canonical and noncanonical signaling pathway of the cytokine TGF-β, recognized as a critical player in the development of fibrosis; even though it was initially described as a marker for activated endothelial cells, ENG is expressed on a wide variety of cell types, e.g., monocytes, macrophages, smooth muscle cells, fibroblasts, neural crest stem cells, and adult bone marrow hematopoietic stem cells [22,23,24,25]. In addition, evidence suggests that ENG is involved in developing some types of cancer, such as hepatocellular carcinoma, and promotes the progression of fibrosis in different organs, e.g., skin, heart, liver, and kidney [22,24,26,27]. Additionally, ENG is associated with cellular processes such as proliferation, migration, adhesion, cytoskeleton organization, and fibroblast activation [22,24,26]. Studies have shown that ENG expression is significantly increased at the mRNA and protein level in fibroblasts from tissues undergoing a fibrotic process than fibroblasts from healthy tissue, which is consistent with the results obtained in our study [22,24,27]. In this regard, it was shown that angiotensin II stimulates the upregulation of ENG at the mRNA and protein level in cardiac fibroblasts and that ENG overexpression correlated with increased collagen production [22,27]. ENG plays an essential role in the fibrotic process of systemic sclerosis (SSc) by promoting the TGF-β signaling pathway through phosphorylation of proteins such as mothers against decapentaplegic homolog 1 (SMAD1) and mothers against decapentaplegic homolog 3 (SMAD3), as well as by inducing the production of ECM-associated proteins such as collagen and connective tissue growth factor (CCN2) [24]. In addition, studies in murine models of unilateral ureteral obstruction (UUO)-induced renal fibrosis demonstrated that ENG-overexpressing mice exhibit a higher degree of fibrosis, accompanied by increased collagen and fibronectin [28]. Therefore, our result indicates that ENG may also participate in IPF fibrogenesis, an intriguing phenomenon that needs to be addressed.

THY1, also called CD90, is a glycoprotein anchored to glycophosphatidylinositol and is expressed on the surface of a wide diversity of cells, including neurons, lymphocytes, endothelial cells, hematopoietic stem cells, and fibroblasts [29,30,31]. In addition, THY1 has been described as a marker of cancer stem cells (CSCs), and its up-regulation has been identified in different types of cancer [32]. Thus, it has been proposed as a cancer biomarker [32,33]. THY1 can activate diverse signaling pathways and fulfill several functions in different tissues, mainly related to cell–cell and cell–ECM interactions; therefore, it influences cellular processes such as proliferation, differentiation, survival, apoptosis, wound healing, and fibrosis [34,35]. The presence of different fibroblast subpopulations that differ in various phenotypic characteristics in the lung and play different roles in fibrotic processes has been described; however, the contribution of these subpopulations in the pathogenesis of IPF is still an unmet need [30,34]. In this regard, it has been reported that THY1 (−) and THY1 (+) fibroblast populations converge in the lung [30].

Findings to date have reported that THY1 (−) lung fibroblasts sustain a higher proliferative, migratory capacity, and resistance to apoptosis than THY1 (+) fibroblasts [29,32,34]. Furthermore, THY1 (−) lung fibroblasts can react better to the profibrotic factors platelet-derived growth factor-BB (PDGF-BB), interleukin-1β (IL-1β), interleukin-4 (IL-4), and bleomycin by inducing activation of latent TGF-β, demonstrating the absence of THY1 expression in fibroblasts correlates with lung fibrogenesis [30]. Other studies in a murine model of bleomycin-induced IPF have shown that Thy-1−/− mice develop more severe pulmonary fibrosis, accompanied by an increase in TGF-β activity and a significant increase in collagen accumulation at 14 days after bleomycin treatment when compared to WT mice [36]. Moreover, THY1 overexpression was recently shown to regulate the progression of acute interstitial pneumonia (AIP) by significantly decreasing the expression of profibrotic proteins such as Matrix Metallopeptidase 2 (MMP-2), Occludin, α-SMA, Vimentin and β-catenin, and phosphorylation levels of β-catenin, resulting in inhibition of the WNT signaling pathway, a critical pathway in the pathogenesis of IPF, resulting in decreased proliferation and increased apoptosis of lung fibroblasts [32].

CD9 is a transmembrane glycoprotein that belongs to the tetraspanin family. It can interact with other tetraspanins or other proteins, such as integrins, growth factor receptors, membrane proteases, immunoglobulins, and intracellular signaling molecules [37,38]. Tetraspanins can dynamically assemble to form functional multiprotein complexes, also called tetraspanin-enriched microdomains that play an essential role in cell–cell and cell–ECM interactions [37,38,39,40]. Therefore, they are closely related to cellular processes such as proliferation, apoptosis, adhesion, migration, signal transduction, and cell differentiation [37,39]. CD9 is ubiquitously expressed on various cells such as monocytes, macrophages, eosinophils, basophils, endothelial cells, epithelial cells, smooth muscle cells, and tumor cell lines [37,39,40]. In addition, CD9 has been associated with various pathologies such as infectious diseases caused by viruses or bacteria, lung inflammation, and cancer [37,38,39,40]. In this context, the critical protective role of CD9 on lung inflammation and emphysema has been reported; for example, CD9 deficiency has been reported to enhance macrophage activation in vitro enormously and aggravate lung inflammation, as well as in vivo, through being induced by lipopolysaccharide (LPS) stimulation [38,40,41]. The proposed mechanism by which CD9 negatively regulates macrophage activation and LPS-induced lung damage is because it prevents the formation of LPS receptor complex, i.e., prevents CD14-dependent receptor assembly on lipid-enriched membrane microdomains [40,41,42]. Therefore, these findings propose that CD9 up-regulation could be a novel therapeutic approach for lung diseases such as IPF and chronic obstructive pulmonary disease (COPD) [40,41,42].

NT5E, also known as CD73, is a glycoprotein coupled to glycosylphosphatidylinositol, ubiquitously expressed in a wide variety of cells and is the main enzyme that catalyzes the formation of extracellular adenosine from the enzymatic hydrolysis of adenosine 5′-monophosphate (AMP) [43,44]. CD73 is actively involved in regulating tissue homeostasis and some pathophysiological processes related to immunity and inflammation; therefore, CD73 has been shown to play an essential role in the pathogenesis of renal, hepatic, and pulmonary fibrosis and some types of cancer [43,44,45,46,47]. For example, in models of liver fibrosis induced by carbon tetrachloride (CCl_4_) or thioacetamide (TAA) in CD73 knockout mice (CD73KO) and wild-type C57BL/6 control mice (WT), it was shown that CD73KO mice developed a lower degree of fibrosis accompanied by a lower collagen content when compared in the livers of WT mice after treatment with CCl_4_ or TAA [45]. Similarly, a model of lung fibrosis induced by a single dose (15 Gray) of chest area irradiation in WT and CD73−/− (Nt5e−/−) mice showed that CD73 plays an essential role in the development of fibrosis and that treatment with radiation leads to a gradual increase in CD73 activity in the lung between 3 and 30 weeks after treatment in WT mice, with a more significant increase observed between 25 and 30 weeks, at which time there was a greater degree of fibrosis. In contrast, CD73−/− mice presented with a lower degree of pulmonary fibrosis [[46]. However, disagreeing with the possible profibrotic effect described above, CD73 is an antifibrotic factor in the context of bleomycin administration-induced pulmonary fibrosis. CD73−/− mice treated with bleomycin for 14 days showed a higher degree of inflammation and fibrosis than WT mice; furthermore, it was observed that fibrosis was attenuated by intranasal administration of exogenous nucleotidase. These results propose that CD73 contributes to anti-inflammatory pathways in bleomycin administration-induced lung fibrosis [48].

ITGA3, also known as integrin α3, is a member of the integrin family, a group of transmembrane proteins composed of an α-chain and a β-chain presenting a sizeable extracellular portion and a small cytoplasmic domain [49,50,51]. In particular, ITGA3 has the faculty to bind to the β1 subunit to form an α3β1 integrin that can interact with various ECM proteins, mainly laminins [49,50,51,52]. Therefore, it plays a crucial role in cell–cell adhesion and cell–ECM adhesion [49,51,52]. In addition, integrin α3β1 is widely expressed in epithelia, especially in the lung, kidney, and skin [49,52]. In this regard, it has been reported that mice are lacking *Itga3* develop severe defects and abnormalities in the epidermis, lungs, and kidneys [49,52]. Moreover, several studies have shown that ITGA3 maintains aberrant expression in various types of cancer, including prostate cancer, intrahepatic cholangiocarcinoma, melanoma, pancreatic cancer, squamous cell carcinoma of the tongue, breast cancer, and colorectal cancer [50,51]. Recently, released information on an *ITGA3* mutation that increases glycosylation of the α3 subunit, results in the inhibition of the binding to the β1 subunit, thus preventing the formation of functional α3β1 integrin, resulting in the manifestation of very severe renal and pulmonary abnormalities [52]. However, the role of ITGA3 in human IPF is still unknown.

Interestingly, our results show that two subunits comprising the non-erythroid spectrin, also known as non-erythroid fodrin (αIIβII), including the αII subunit (SPTAN1,) also called αII-spectrin and which is known to be the non-erythroid homolog of αI-spectrin, more commonly referred to as α-fodrin, and the βII subunit (SPTBN1), more commonly referred to as β-fodrin, were up-regulated in lung fibroblast cell lines with IPF phenotype when compared to normal lung fibroblasts [53]. SPTAN1 and SPTBN1 are cytoskeleton proteins that are part of the spectrin family. These include several structural proteins such as α-actinin, dystrophin, and utrophin, whose function is essential for the maintenance of the structural integrity of the cytoskeleton and proper cellular function [53,54,55]. Therefore, they fulfill various functions associated with cellular mechanisms such as cell adhesion, cell cycle, angiogenesis, apoptosis, and the epithelial–mesenchymal transition (EMT) process [53,54]. In this regard, the role of SPTAN1 and SPTBN1 in pathologies such as cancer has been reported [53,54,56]. However, in the case of SPTAN1, evidence has suggested that it could act as a tumor suppressor in some types of cancer, such as prostate cancer, lung cancer, and colorectal cancer. In contrast, in colorectal cancer, breast cancer, gastric cancer, and lung cancer, it has been shown that it could act as a tumor promoter [53,54]. In the case of SPTBN1, data support that its expression levels are significantly down-regulated in various human cancers such as digestive tract cancer, pancreatic cancer, hepatocellular carcinoma, and lung cancer, reducing SPTBN1 expression has been linked to cancer progression [53,56]. This suggests that SPTBN1 may act as a tumor inhibitor [56]. However, to date, there is no evidence of the biological role played by SPTAN1 and SPTBN1 in the development of IPF.

ALCAM, also known as CD166, an adhesion protein belonging to the immunoglobulin superfamily, is expressed on endothelial cells, epithelial cells, lymphoid cells, myeloid cells, neuronal cells, bone marrow cells, fibroblasts, and hepatocytes [57,58]. In addition, CD166 has been reported as a possible marker of CSCs in different types of cancer, highlighting its potential cancer-promoting role [57,59,60]. Moreover, CD166 is involved in cellular processes such as proliferation, migration, invasion, adhesion, hematopoiesis, immune response, neurogenesis, and cancer metastasis [57,59,61]. For example, down-regulation of CD166 in a nasopharyngeal carcinoma-derived epithelial cell line (CNE-2R cells) significantly attenuated proliferation, invasion, and the EMT processes [62]. However, studies on CD166 and its relationship to IPF development are limited.

ANPEP or CD13 is a multifunctional transmembrane protease initially identified on the cell surface of myeloid cells. Still, different cell types such as fibroblasts, endothelial cells, synoviocytes, epithelial cells, and pericytes have been shown to express ANPEP ubiquitously [63,64,65]. In addition, CD13 was identified as a marker capable of differentiating two native clonogenic mesenchymal cell populations in the human lung that show unequal proliferative capacity [66]. This protein has been implicated in the pathogenesis of lung diseases such as acute respiratory distress syndrome (ARDS) and various autoimmune disorders such as scleroderma, rheumatoid arthritis, and psoriasis [63,65,67,68]. ANPEP can act as an enzyme and cleave the N-terminal regions of multiple cytokines and regulate their activity; also, it is involved in antigen processing by cleaving major histocompatibility complex class (MHC) II-associated peptides present on the surface of antigen-presenting cells [64,68]. In addition, it can act as a cell surface receptor and participate in some cellular functions such as proliferation, migration, invasion, adhesion, apoptosis, angiogenesis, and antigen processing [63,64,67]. Recently, ANPEP has been described as a viral receptor of human coronavirus [67,69,70]. However, its involvement in the pathogenesis of IPF has not been described.

ARFGAP1 is a protein that is part of the ArfGAP family of regulators essential for membrane trafficking pathways [71,72]. ARFGAP1 localizes mainly to the cytosolic side of Golgi cisternal membranes, and its primary functions are related to the formation of clathrin-coated vesicles and (coat complex protein I) COPI [71,73]. Recently, it was described that ARFGAP1 might act as a critical regulator of mechanistic target of rapamycin complex 1 (mTORC1) by inhibiting lysosomal localization and activation of mTORC1, resulting in decreased cell growth; therefore, these results propose its potential for cancer therapy [74]. However, its involvement in the pathogenesis of IPF has not been described.

By virtue of the results obtained, we can suggest that this signature of DEPs provides relevant information that contributes to improving the understanding of the phenotypic changes observed in IPF fibroblasts. Furthermore, with the available information on the biological function of these DEPs, we can hypothesize that they may be actively participating in different cellular processes related to the promotion of IPF, such as cell proliferation, migration, invasion, adhesion, survival, differentiation, cytoskeleton organization, and fibroblast activation. Equally important, the results shown provide initial information on these DEPs and their possible clinical application as potential new therapeutic targets to prevent the development of IPF and as candidate biomarkers of the disease. However, the present study has several limitations. First, additional experimental evidence is needed to validate the involvement of these DEPs in IPF-related cellular and molecular processes involving fibroblasts. Second, given that we employed an in vitro model of IPF, future studies are required to validate the role of these DEPs in the pathogenesis of IPF and their potential clinical applications as therapeutic targets and biomarkers for in vivo models of IPF and samples from patients with IPF.

## 4. Materials and Methods

### 4.1. Cell Culture

Normal (CCD19Lu) and those bearing a human IPF (LL29 and LL97A) lung fibroblast cell lines were purchased from the American Type Culture Collection (ATCC, Cat. No. CCL-210; No. CCL-134 and No. CCL-191, respectively; Manassas, VA, USA). Cell lines were cultured in Eagle’s minimal essential medium (EMEM) (ATCC, Manassas, VA, USA) supplemented with 10% fetal bovine serum (SFB) (Gibco, Thermo Fisher Scientific, Waltham, MA, USA) and 1% penicillin/streptomycin (Gibco, Thermo Fisher Scientific, Waltham, MA, USA). Maintained cells in a humidified incubation at 37 °C and 5% CO_2_, and experiments were performed using passages from 8 to 12.

### 4.2. Protein Extraction and Quantification

CCD19Lu (NL-1), LL29 (IPF-1), and LL97A (IPF-2) cell lines were grown in 10 cm culture plates to 70% confluence. Cells were then washed three times with PBS before adding a fresh culture medium supplemented with 5% SFB. After incubation for 48 h (85–95% confluency), cells were scraped from the culture plates, followed by centrifugation to create a cell pellet. Finally, cell pellets were resuspended in an 8 M urea lysis buffer for label-free proteomic analysis. Protein concentration was measured by a bicinchoninic acid (BCA) assay (Thermo Fisher Scientific, Waltham, MA, USA).

### 4.3. Label-Free Quantitative Proteomic Analysis

#### 4.3.1. Sample Preparation for Proteomic Analysis

The total protein lysate was centrifuged at 16,000× *g*, 4 °C for 15 min, and the supernatants were transferred to a clean Eppendorf tube. Subsequently, proteins were precipitated using cold acetone. Then, it was centrifuged at 16,000× *g*, 4 °C for 15 min, and the supernatant was discarded. Next, the protein pellets were dissolved in a 6 M aqueous urea solution, and 30 μg of total protein was denatured with 10 mM DL-dithiothreitol (DTT) (Sigma-Aldrich, St. Louis, MO, USA) by incubating it at 56 ℃ for 1 h, followed by alkylation with 50 mM iodoacetamide (IAA) (Sigma-Aldrich, St. Louis, MO, USA) by incubating for 60 min at room temperature in the dark. Then, 500 mM ammonium bicarbonate was added into the solution to make a final concentration of 50 mM ammonium bicarbonate with a pH of 7.8. Trypsin (Promega, Madison, WI, USA) was added to the protein solution for digestion at 37 °C for 15 h. The generated peptides were further purified with ZipTip to remove the salt. Finally, the samples were dried under vacuum and stored at −20 °C until use.

#### 4.3.2. LC-MS/MS Analysis

All LC-MS/MS analyses were performed using an Ultimate 3000 nano UHPLC system coupled to a Q Exactive HF mass spectrometer (Thermo Fisher Scientific, Waltham, MA, USA) and an ESI nanospray source. Three comparative groups were analyzed, meaning a total of 6 samples, i.e., two biological replicates per group. A total of 1 μg of peptide sample was separated in a two-column configuration with a capture column (PepMap C18, 100 Å, 100 μm × 2 cm, 5 μm), followed by an analytical column (PepMap C18, 100 Å, 75 μm × 50 cm, 2 μm). Separations were achieved using a gradient of (A) 0.1% formic acid (FA) (Sigma-Aldrich, St. Louis, MO, USA) in water and (B) 0.1% FA in 80% acetonitrile (CAN) (Sigma-Aldrich, St. Louis, MO, USA). The gradient conditions were as follows: 2 to 8% buffer B for 3 min, followed by an increase from 8 to 20% buffer B for 56 min. Next, a gradient of 20 to 40% buffer B was used for 37 min, followed by a 40 to 90% buffer B for 4 min at a 250 nL/min flow rate. Full scans were acquired at a resolution setting of 60,000–200 *m/z*, with a scan range of 300 and 1650 *m/z*, using an automatic gain control (AGC) target of 3.0 × 10^6^. MS/MS scans were acquired in Top 20 mode with a resolution setting of 15,000–200 *m/z* with an AGC target of 1.0 × 10^5^ and a maximum injection time 19 ms. We normalized the collision energy to 28%; 1.4 Th isolation window; charge states were excluded; unassigned, 1, >6. Set dynamic exclusion time at 30 s.

#### 4.3.3. Proteomics Data Analysis

The raw files were analyzed using Maxquant (v.1.6.2.6) (Max Planck Institute, Martinsried, Germany). The MS/MS spectra were searched against the species-based human protein database of the sample used. For protein identification analysis and LFQ, we established the following parameters:Enzyme specificity was set to trypsin.Maximum missed cleavages were set to 2.Precursor ion mass tolerance was set to 10 ppm, and MS/MS tolerance was 0.6 Da.Protein modifications included carbamidomethylation (C) (fixed), oxidation (M) (variable).The false discovery rate (FDR) of peptides and proteins was 0.01.

Proteins were quantified using LFQ intensity, and removed proteins without MS/MS, and LFQ intensity. Proteins with LFQ ≠ 0 in at least 1 out of 6 samples (2 biological replicates per group) were retained for further statistical analysis using Perseus v1.6.15.0 (Max Planck Institute, Martinsried, Germany) [75]. For quantitative analysis of the LFQ data, we performed a log2 transformation of the LFQ values. Then, we filtered the results to remove possible contaminants, reverse matches, and uniquely identified proteins per site. Additionally, we filtered them to retain proteins with an MS/MS spectral count ≥ 2. The data were then row filtered according to valid values (minimum valid percentage, 70%). Then, they were normalized by median subtraction. Further imputation of missing values was performed by selecting a downward shift of 1.8 and a width of 0.3 standard deviations in a normal distribution to simulate low abundance protein signals. Finally, two-tailed Student’s t-tests were performed for pairwise comparisons of proteomes, i.e., IPF-1 vs. NL-1, IPF-2 vs. NL-1, to detect DEPs Therefore, a corrected *p*-value of 0.05 with a FC ≥ 1.5 and ≤ −1.5 was set as the cutoff point to determine whether a protein was differentially expressed. We developed a 3-way Venn diagram to observe the distribution of proteins identified by proteomic analysis using the interactiVenn web application (http://www.interactivenn.net (accessed on 8 December 2021)) [76]. Protein identification data were further processed and visualized using GraphPad Prism 8 (San Diego, CA, USA) and RStudio (Boston, MA, USA) statistical software.

### 4.4. Bioinformatics Analysis

The GO functional annotations, including BP, MF, and CC, and KEGG pathway analysis, were obtained using the David 6.8 database (https://david.ncifcrf.gov (accessed on 15 January 2022)) [77]. PPI networks were constructed using STRING v.11.5 (https://string-db.org (accessed on 8 December 2021)), the network was constructed with a minimum required interaction score > 0.4. Cytoscape v.3.8.2 software (Cytoscape Consortium, San Diego, CA, USA) was used to visualize the network [78,79]. In addition, the Cytoscape add-on CytoHubba was used to explore the hub proteins of the PPI network using the MCC method [80].

### 4.5. Statistical Analysis

We used the Student’s *t*-test for unpaired, two-tailed data to determine the statistical significance of FC in log2-transformed proteomic data.

## 5. Conclusions

In conclusion, using label-free quantitative proteomics, we revealed a novel signature of DEPs in lung fibroblast cell lines with IPF phenotype when compared to normal lung fibroblasts. Among these, essential proteins such as ENG, THY1, CD9, NT5E, ITGA3, SPTBN1, SPTAN1, ALCAM, ANPEP, and ARFGAP1 were identified and are highlighted because they might be involved in the pathogenesis of human IPF. Furthermore, the presented results provide preliminary information on these DEPs and their possible application for discovering new potential therapeutic targets to halt the development of IPF and for the identification of candidate biomarkers of the disease. Therefore, this study warrants further studies to comprehensively and conclusively elucidate the role of these proteins in the pathogenesis of IPF and their potential clinical applications.

## Figures and Tables

**Figure 1 ijms-23-05032-f001:**
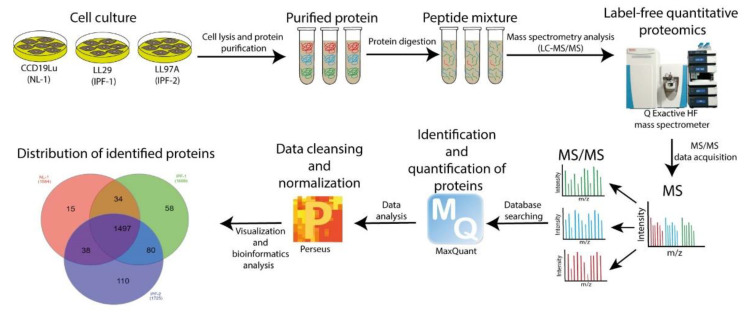
Schematic representation of the experimental design and workflow of label-free proteomic analysis. Cell lines are subjected to cell lysis and processed to extract proteins. These are digested and analyzed by LC-MS/MS. Peptides are identified and quantified by label-free methods.

**Figure 2 ijms-23-05032-f002:**
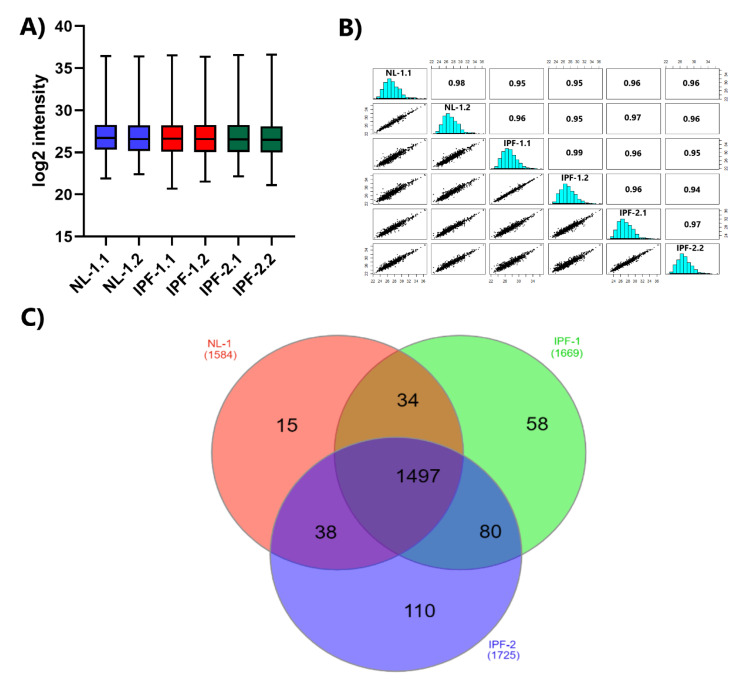
LC-MS/MS reproducibility and distribution of proteins identified. (**A**) Box plot of biological replicates of each cell line. (**B**) Correlation plot of biological replicates of all cell lines. (**C**) Venn diagram showing the distribution of proteins in the different cell lines. IPF, idiopathic pulmonary fibrosis; NL, normal lung.

**Figure 3 ijms-23-05032-f003:**
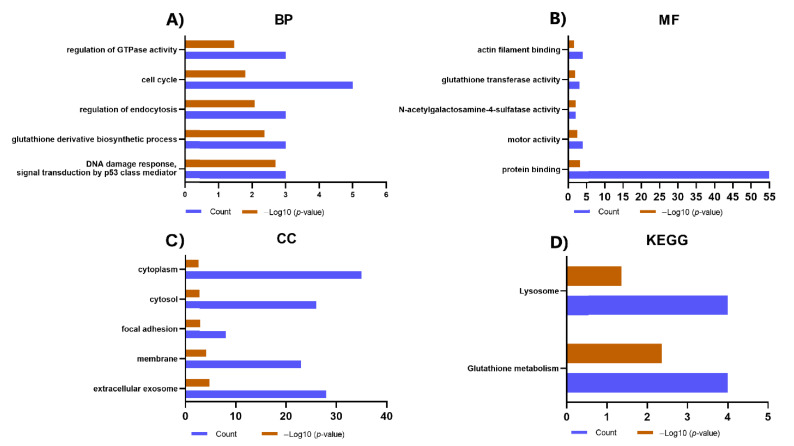
GO and KEGG analysis of proteins identified exclusively in the IPF-2 and IPF-1 groups. (**A**) The main enriched terms for BP. (**B**) The main terms enriched for MF. (**C**) The main enriched terms for CC. (**D**) Top enriched terms for KEGG pathways. If there were more than five enriched terms for one of the categories, the five most representative ones were selected according to the *p*-value.

**Figure 4 ijms-23-05032-f004:**
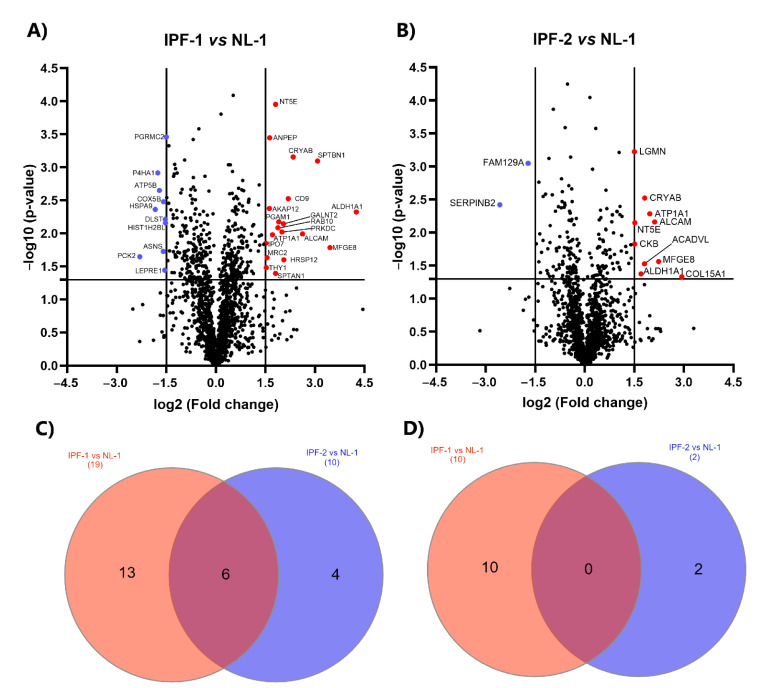
Differentially expressed proteins. (**A**,**B**) Volcano plot showing both up-regulated (red) and down-regulated (blue) proteins in all pairwise comparisons. Volcano plots depict FC (*x*-axis) and −log10 value of *p*-value (*y*-axis). Red dots in the upper right (ratio ≥ 1.5) and blue dots in the upper left (ratio ≤ 1.5) sections represent significantly deregulated proteins, *p* < 0.05. (**C**) Venn diagram showing the distribution of up-regulated proteins in the comparison groups. (**D**) Venn diagram showing the distribution of down-regulated proteins in the comparison groups. IPF, idiopathic pulmonary fibrosis; NL, normal lung.

**Figure 5 ijms-23-05032-f005:**
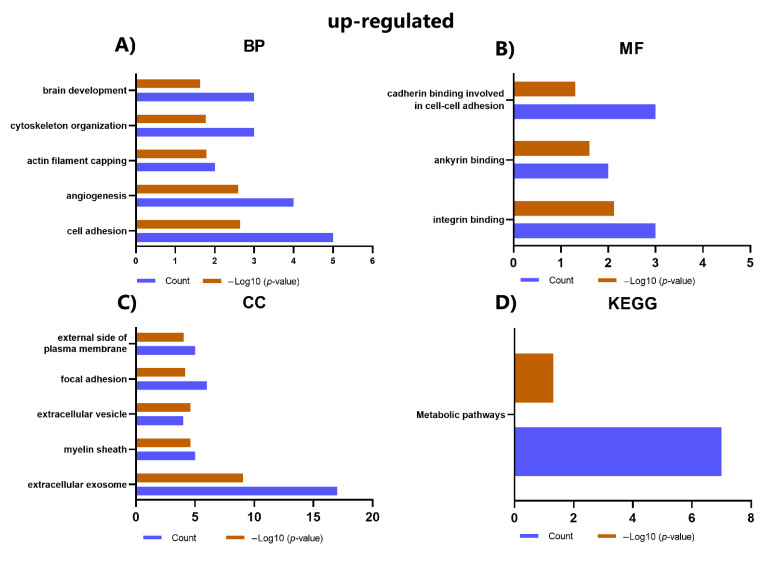
GO and KEGG analysis of DEPs (**A**) The significant terms enriched for BP in up-regulated proteins. (**B**) The significant terms enriched for MF in the up-regulated proteins. (**C**) The main terms enriched for CC in up-regulated proteins. (**D**) Top enriched terms for KEGG pathways in up-regulated proteins. (**E**) The significantly enriched terms for BP in the down-regulated proteins. (**F**) Top enriched terms for CC in down-regulated proteins. (**G**) Top enriched terms for KEGG pathways in down-regulated proteins. If there were more than five enriched terms for one of the categories, the five most representative ones were selected according to the *p*-value.

**Figure 6 ijms-23-05032-f006:**
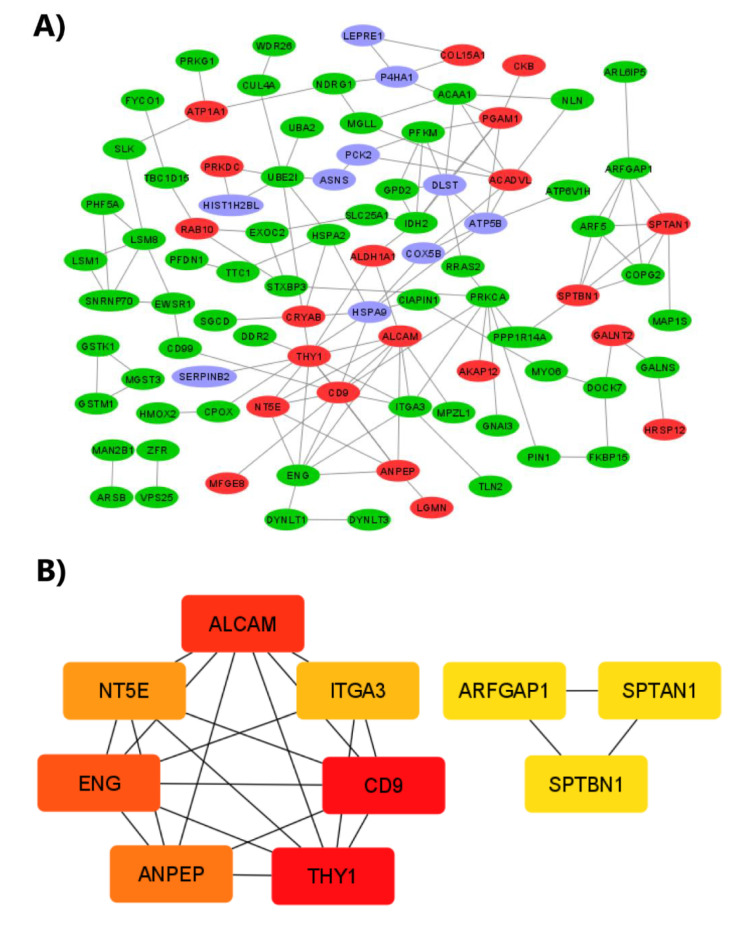
Protein–protein interaction regulatory network. (**A**) DEPs and proteins identified exclusively in the IPF-1 and IPF-2 groups were combined to construct a regulatory network using STRING software to visualize the interaction with evidence such as network edge importance. The active interaction sources were Text Extraction, Experiments, Database, Co-expression, Neighborhood, Gene Fusion, and Co-occurrence, with a required minimum interaction score of medium confidence (0.4). The red color represents up-regulated proteins, the blue color represents down-regulated proteins, and the green color represents proteins identified exclusively in the IPF-1 and IPF-2 groups. (**B**) The top 10 proteins with the highest degree of PPI network connectivity were identified by the MMC method using CytoHubba.

**Table 1 ijms-23-05032-t001:** GO and KEGG analysis of the 10 hub proteins.

Category	Term	Count	*p*-Value	Proteins
BP	GO:0007155~cell adhesion	5	6.23 × 10^5^	ALCAM, ITGA3, CD9, THY1, ENG
BP	GO:0043547~positive regulation of GTPase activity	4	0.00273472	THY1, SPTAN1, SPTBN1, ARFGAP1
BP	GO:0006888~ER to Golgi vesicle-mediated transport	3	0.0031085	SPTAN1, SPTBN1, ARFGAP1
BP	GO:0007010~cytoskeleton organization	3	0.00314673	THY1, SPTAN1, SPTBN1
BP	GO:0051693~actin filament capping	2	0.00694772	SPTAN1, SPTBN1
BP	GO:0017015~regulation of transforming growth factor-beta receptor signaling pathway	2	0.01067099	ITGA3, ENG
BP	GO:0030336~negative regulation of cell migration	2	0.04979153	THY1, ENG
MF	GO:0005178~integrin binding	3	0.00134085	ITGA3, CD9, THY1
CC	GO:0009897~external side of plasma membrane	6	2.52 × 10^8^	ALCAM, ITGA3, ANPEP, CD9, THY1, ENG
CC	GO:0005925~focal adhesion	5	2.41 × 10^5^	ALCAM, ITGA3, CD9, THY1, ENG
CC	GO:0070062~extracellular exosome	8	5.57 × 10^5^	NT5E, ALCAM, ITGA3, ANPEP, CD9, THY1, SPTAN1, SPTBN1
CC	GO:0008091~spectrin	2	0.00443689	SPTAN1, SPTBN1
CC	GO:1903561~extracellular vesicle	2	0.02442875	CD9, SPTAN1
CC	GO:0005887~integral component of plasma membrane	4	0.0275047	ALCAM, ANPEP, CD9, THY1
CC	GO:0009986~cell surface	3	0.02767149	NT5E, ITGA3, ENG
KEGG	hsa04640: Hematopoietic cell lineage	3	0.00229492	ITGA3, ANPEP, CD9

## Data Availability

Data presented in this study are available on request from the corresponding author.

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
