# Peer review of "Proteomic Analysis Reveals Differential Expression Profiles in Idiopathic Pulmonary Fibrosis Cell Lines"

_ijms, 2022, doi:10.3390/ijms23095032_

Round 1
Reviewer 1 Report
- the abstract does not respect the IMRAD structure.
- The terms utilised as abbreviation must be explained for the first time when are used.
- in the introduction section the aim is not well established.
- the article is difficult to read and maybe a clinician doctor will not find the clinical outcome of your research.
- in the result section you specified table s1 which os not found in the article (line 83).
- the material and methods section must be before the results section.
- in the discussion section you resume again the results without additional explanations and without extrapolation to human pathogenesis and clinical aplicability.
- the conclusions are too general, without a clear outcome.
Author Response
Reviewer # 1
First of all, the authors thank the reviewer for his valuable comments and suggestions that improved the quality of the article. Responses to specific comments are given below.
- The abstract does not respect the IMRAD structure.
Answer: Thank you very much for this detailed comment. We have corrected the Abstract section in the revised version of our manuscript according to the IMRAD structure as shown below.
Introduction: Idiopathic pulmonary fibrosis (IPF) is a chronic, progressive, irreversible lung disorder of unknown cause. This disease is characterized by profibrotic activation of resident pulmonary fibroblasts resulting in aberrant deposition of extracellular matrix (ECM) proteins. However, although much is known about the pathophysiology of IPF, the cellular and molecular processes that occur and allow aberrant fibroblast activation remain an unmet need.
Methods: To explore the differentially expressed proteins (DEPs) associated with aberrant activation of these fibroblasts, we used the IPF lung fibroblast cell lines LL97A (IPF-1) and LL29 (IPF-2) compared to the normal lung fibroblast cell line CCD19Lu (NL -1). Protein samples were quantified and identified using a label-free quantitative proteomic analysis approach by LC-MS/MS. DEPs were identified after pairwise comparison, including all experimental groups. GO enrichment analysis, KEGG, and PPI network construction were used to interpret the proteomic data.
Results: Eighty proteins expressed exclusively in the IPF-1 and IPF-2 clusters were identified. In addition, 19 proteins were identified up-regulated in IPF-1 and 10 in IPF-2; 10 proteins down-regulated in IPF-1 and 2 in IPF-2 compared to the NL-1 proteome. Using STRING software, a PPI network was constructed between the DEPs and the 80 proteins expressed exclusively in the IPF-2 and IPF-1 clusters, containing 115 nodes and 136 edges. The 10 hub proteins present in the IPP network were identified using the CytoHubba plugin of the Cytoscape software. GO and KEGG pathway analyses showed that the core proteins were mainly related to cell adhesion, integrin binding, and hematopoietic cell lineage.
Discussion/Conclusion: Our results provide relevant information on DEPs present in IPF lung fibroblast cell lines compared to the normal lung fibroblast cell line that could play a key role during IPF pathogenesis.
With this important comment in mind, it is essential to mention that the abstract section is included in the manuscript as a single paragraph according to the IMRAD structure, but without headings: 1) Background, 2) Methods, 3) Results, 4) Conclusion. Because the guidelines for authors outlined by the International Journal of Molecular Sciences suggest it this way, they maintain this established format for all articles submitted for peer review.
- The terms utilised as abbreviation must be explained for the first time when are used.
Answer: Thank you very much for this constructive observation. In the revised version of our manuscript, we had now described the terms as used as abbreviations when they were first mentioned in the manuscript.
- In the introduction section the aim is not well established.
Answer: Thank you very much for your comment which helps to improve our manuscript. In the revised version of our manuscript, we have now improved the wording of the paragraph describing our research objectives.
The corrected paragraph has been included in the introduction section, on page #2, lines #73 to #92, as follows:
In the last decade, proteomic tools have enabled high-throughput studies to identify protein expression in different chronic diseases [13,14]. Therefore, the use of proteomics has facilitated the identification of specific diagnostic biomarkers, new therapeutic targets, and critical proteins related to the main molecular mechanisms responsible for developing a specific disease [13,15]. In this regard, a significant number of investigations have focused on the application of proteomic approaches to study the differential proteomic profile in different samples from IPF patients, such as lung tissue, serum, and bronchoalveolar lavage fluid (BALF) [15-20]; however, limited studies have aimed to identify the proteomic profile of IPF fibroblast cell lines.
In the present study, we performed a label-free quantitative proteomic analysis by liquid chromatography-tandem mass spectrometry (LC-MS/MS) of IPF lung fibroblasts compared to normal lung fibroblasts (NL) to gain insight into the profiles of differentially expressed proteins (DEPs). Furthermore, we employed a bioinformatics approach to describe the pathways, biological processes (BPs), molecular functions (MFs), and cellular components (CCs) related to these proteins that might be involved in the pathogenesis of IPF. Therefore, this work aimed to identify DEPs and provide information on the various cellular and molecular pathways activated in IPF fibroblasts. These results provide relevant information that would contribute to the discovery of new potential therapeutic targets to stop the development of IPF, and candidate biomarkers of the disease, and provide a foundation that helps to understand fibroblasts role in the development of IPF.
- The article is difficult to read and maybe a clinician doctor will not find the clinical outcome of your research.
Answer: We appreciate the reviewer's suggestion. With this important comment in mind, we have completely revised the manuscript in accordance with the reviewers' criticisms. We have carefully incorporated the reviewers' recommended modifications throughout the text, mainly in the abstract, introduction, discussion, and conclusion sections. Needless to say, the quality of the revised manuscript has been considerably improved by the suggested recommendations. In addition, we have added relevant information on important points associated with the possible clinical application of the results obtained in the research.
- In the result section you specified table s1 which os not found in the article (line 83).
Answer: We sincerely appreciate this observation and apologize for the omission. Perhaps we made a mistake when uploading the manuscript and supplementary material to the journal platform. Therefore, we re-share the Excel document containing the supplementary tables and also share the Word document containing the supplementary figures:
Table S1: Proteins identified by proteomic analysis using LFQ intensity value ≠ 0 and MS/MS (spectral) count ≥2 in at least one of the biological replicates.
Table S2: Proteins identified in a single group.
Table S3: Proteins identified exclusively in the IPF-1 and IPF-2 groups
Table S4: Proteins identified as differentially expressed (fold change ≥1.5 y ≤ -1.5 and p < 0.05) in IPF-1 vs. NL-1 comparison.
Table S5: Proteins identified as differentially expressed (fold change ≥1.5 y ≤ -1.5 and p < 0.05) in IPF-2 vs. NL-1 comparison.
Figure S1. PPI network of the 80 proteins identified exclusively in the fibrotic groups (IPF-1 and IPF-2)
Figure S2. PPI network of DEPs in the fibrotic groups (IPF-1 and IPF-2).
- The material and methods section must be before the results section.
Answer: We appreciate the reviewer's suggestion. With this important comment in mind, it is essential to mention that the materials and methods section is included in the manuscript after the results section because the guidelines for authors outlined by the international journal of molecular sciences suggest it in this way, and they maintain this established format for all articles submitted for peer review.
- In the discussion section you resume again the results without additional explanations and without extrapolation to human pathogenesis and clinical aplicability.
Answer: We agree with this important suggestion. In the revised version of our manuscript, we have now improved the wording of the paragraph describing the most important findings presented in our research, emphasizing the relationship of DEPs with their possible involvement in different cellular processes associated with IPF and their possible clinical application in future studies.
The corrected paragraph has been included in the discussion section, on page 14, lines #439 to #453, as follows:
By virtue of the results obtained, we can suggest that this signature of DEPs pro-vides relevant information that contributes to improving the understanding of the phenotypic changes observed in IPF fibroblasts. Furthermore, with the available information on the biological function of these DEPs, we can hypothesize that they may be actively participating in different cellular processes related to the promotion of IPF, such as cell proliferation, migration, invasion, adhesion, survival, differentiation, cytoskeleton organization, and fibroblast activation. Equally important, the results shown provide initial information on these DEPs and their possible clinical application as potential new therapeutic targets to prevent the development of IPF and as candidate biomarkers of the disease. However, the present study has several limitations. First, additional experimental evidence is needed to validate the involvement of these DEPs in IPF-related cellular and molecular processes involving fibroblasts. Second, given that we employed an in vitro model of IPF, future studies are required to validate the role of these DEPs in the pathogenesis of IPF and their potential clinical applications as therapeutic targets and biomarkers in vivo models of IPF and samples from patients with IPF.
- The conclusions are too general, without a clear outcome
Answer: Thank you very much for this constructive comment. In the revised version of our manuscript, we have improved the wording of the conclusions of our manuscript.
The corrected paragraph has been included in the conclusion section, on page 16, lines #547 to #556, as follows:
In conclusion, using label-free quantitative proteomics, we revealed a novel signature of DEPs in lung fibroblast cell lines with IPF phenotype compared to normal lung fibroblasts. Among these, essential proteins such as ENG, THY1, CD9, NT5E, ITGA3, SPTBN1, SPTAN1, ALCAM, ANPEP, and ARFGAP1 were identified and are highlighted because they might be involved in the pathogenesis of human IPF. Furthermore, the presented results provide preliminary information on these DEPs and their possible application for discovering new potential therapeutic targets to halt the development of IPF and for the identification of candidate biomarkers of the disease. Therefore, this study warrants further studies to comprehensively and conclusively elucidate the role of these proteins in the pathogenesis of IPF and their potential clinical applications.
Finally, we are very grateful for the relevant suggestions/comments of Reviewer #1 and hope that the changes in the revised version of our manuscript are satisfactory and acceptable for publication in the international journal of molecular sciences.

Reviewer 2 Report
This study used label-free quantitative proteomic analysis by LC-MS/MS with a bioinformatics approach to identify differentially expressed proteins (DEPs) and their interactions in IPF lung fibroblasts compared with normal lung fibroblasts. The authors observed that proteome analysis of IPF lung fibroblast cell lines, compared with the normal lung fibroblast cell line, increases the understanding of the pathophysiological changes associated with IPF.
The manuscript is well written and organized. The experimental design has been well conceived and carefully carried out. Methods are appropriate, results are clearly described and illustrated, as well as properly discussed. References are quite relevant and updated. This paper can be useful for the International Journal of Molecular Sciences readers, because it provides very interesting information within the current context of only few published studies.
Author Response
Reviewer # 2
This study used label-free quantitative proteomic analysis by LC-MS/MS with a bioinformatics approach to identify differentially expressed proteins (DEPs) and their interactions in IPF lung fibroblasts compared with normal lung fibroblasts. The authors observed that proteome analysis of IPF lung fibroblast cell lines, compared with the normal lung fibroblast cell line, increases the understanding of the pathophysiological changes associated with IPF.
The manuscript is well written and organized. The experimental design has been well conceived and carefully carried out. Methods are appropriate, results are clearly described and illustrated, as well as properly discussed. References are quite relevant and updated. This paper can be useful for the International Journal of Molecular Sciences readers, because it provides very interesting information within the current context of only few published studies.
Answer: Thank you very much for the encouraging comment about our investigation. We hope that our report contributes to the underlying molecular mechanisms associated with IPF pathogenesis. Thank you very much for helping us to improve our manuscript!

Round 2
Reviewer 1 Report
Thank you for your answers and your article revision.